# Crisis Preparedness Exercise on Rift Valley Fever Introduction into Europe under a One Health Approach

**DOI:** 10.3390/microorganisms10091864

**Published:** 2022-09-18

**Authors:** Ombretta Pediconi, Silvia D’Albenzio, Georgia Gkrintzali, Paolo Calistri, Milen Georgiev

**Affiliations:** 1Istituto Zooprofilattico Sperimentale dell’Abruzzo e del Molise “G. Caporale”, 64100 Teramo, Italy; 2European Food Safety Authority, 43126 Parma, Italy

**Keywords:** Rift Valley Fever, simulation exercise, emergency preparedness, One Health

## Abstract

Crisis preparedness training programmes are substantial for the effective management of contingency plans. Rift Valley Fever (RVF) was chosen as the vector transmitted zoonosis for a crisis preparedness exercise co-organised in 2021 by the European Food Safety Authority (EFSA) and the Istituto Zooprofilattico Sperimentale dell’Abruzzo e del Molise “G. Caporale” (IZS-Teramo). The online table-top simulation exercise was planned to strengthen the network of Mediterranean countries on rapid risk assessment, risk/crisis management and risk communication during a human/animal health crisis, adopting the ‘One Health’ approach. Italy, Spain, Portugal, France, Greece, Albania, Croatia, Montenegro and Turkey were the beneficiary countries, while European Commission (EC), European Centre for Disease Prevention and Control (ECDC), World Health Organisation (WHO), World Organisation for Animal Health (WOAH) and Food and Agricultural Organisation (FAO) were the designated observers who were actively involved along the entire capacity building process. The simulation exercise was based on a fictional case study in which the zoonotic mosquito-borne disease, not currently present in Europe, was accidentally introduced into the European Union via the accidental transfer of infected vectors from a RVF-endemic country. The training activity was positively assessed by the participants and useful suggestions were given to address further future similar initiatives.

## 1. Introduction

Crisis preparedness training programmes are substantial for the effective management of contingency. They are crucial elements in preparedness schemes that contribute to the efficient and timely analysis and response to threats, including those directed to human, animal, plant health, food safety or food supply chain, and with the potential to affect different countries and various sub-populations in time.

The European Food Safety Authority (EFSA) is the European Risk Assessment Agency in the areas of food, feed safety, plant, animal health and welfare. In the field of crisis preparedness and response, through multiannual programmes of training and workshops, EFSA enhances and intensifies the collaboration with its institutional stakeholders to be prepared for various crisis scenarios.

General Food Law and specifically article 55 of Council Regulation (EC) No. 178/2002 [1] outlines the responsibilities of the European Commission (EC) to draw up, in close cooperation with EFSA and the Member States (MSs) of the European Union (EU), a general plan for crisis management in the field of food and feed safety. According to the Commission Implementing Decision (EU) 2019/3002 [2], this plan is activated when a situation that presents a risk to consumers, animal, or plant health and cannot be controlled by the existing mechanisms. EFSA’s role in the area of a food/feed safety crisis is to provide scientific and technical assistance in the crisis management procedures and be able to respond quickly and efficiently. In this framework, EFSA has drawn up its internal procedures for situations requiring urgent responses, which have been updated in 2021 [3].

In this framework, EFSA has organised several crisis preparedness exercises and activities in the past to plan and practise how to respond to possible emergencies, including several workshop-based exercises between 2012 and 2022 [4,5,6,7,8,9,10,11,12].

According to EFSA’s multiannual (2021–2024) crisis preparedness training strategy [13], whose theme is ‘Trusted Transparent Response’, which means not only ‘doing the right thing, collaboratively’ but also ‘being perceived to have done the right thing in a very open and transparent way’, EFSA co-organised in 2021 a crisis preparedness exercise with Istituto Zooprofilattico Sperimentale dell’Abruzzo e del Molise “G. Caporale” (IZS-Teramo). The exercise was focused on a vector transmitted zoonosis, Rift Valley Fever (RVF) with participants from several Mediterranean countries and representatives of international organisations.

The general aim of the event was ‘To strengthen the network of Italy and its Mediterranean neighbouring countries on rapid risk assessment, risk/crisis management and risk communication during a human health/animal health crisis, adopting the “One Health” approach’. The objective of the training was the elaboration of ways to improve the incidence response collaboration between animal and public health authorities from multiple countries, using a “One Health’ approach” as well as the improvement of skills in the areas of outbreak investigations, control of RVF outbreak, identification of preventive actions for RVF and the effective relevant communication to stakeholders.

## 2. Materials and Methods

This simulation exercise was based on a case study considering the introduction of a zoonotic mosquito-borne disease, such as RVF, not currently present in Europe, through the accidental introduction of infected vectors from a RVF-endemic country.

In particular, a specific scenario was presented to the participants of the event, which considered the arrival to Croatian ports in late July of a sea cargo originating from Port Sudan (Sudan), bringing several containers of various materials, including orchid bulbs and other tropical flowers with soil. In late August, the first clinical signs (abortions and mortality in lambs) were observed in some sheep farms in the proximity of a Croatian port, later diagnosticated as cases of RVF.

The hypothesis of the introduction of vectors infected by the RVF virus (RVFV) through sea cargoes is in line with the possible scenarios on RVFV introduction into Europe, according to the EFSA opinion on the risk of introduction of RVFV into the EU, adopted in January 2020 [14].

The choice of the introduction into the Croatian coast was made for training purposes only, and no considerations were made to the specific national authorities or institutions of any country.

The scenario considered three chronologically distinct phases, during which crucial decisions must be taken:(1)Initially, after the confirmation of RVF presence, in which early response activities must be organised,(2)Later, when surveillance actions must be designed and implemented for a proper assessment of the epidemiological situation; and(3)Finally, when control strategies must be identified, in response to the establishment of the disease in part of the territory.

The table-top simulation exercise was implemented immediately after a one-month preparatory eLearning training phase aimed to provide the participants with the necessary basic-level knowledge of RVF. An original online eLearning pathway on RVF, totally self-supportive, has been produced by IZS-Teramo to present the main morphological characteristics of the virus, its pathogenesis and the main epidemiological features, with specific reference to the geographic distribution, transmission, receptive hosts, reservoirs, vectors and the factors that favour its spread. This online course was divided into six sections and included a detailed review of the clinical signs and the anatomo-histopathological lesions caused by RVF in sheep, goats, cattle, camels and humans, the main diagnostic tools available and the possible monitoring and controlling strategies to adopt for limiting the RVF spread.

Crucial was the multi-disciplinary dimension of the team of experts engaged by IZS-Teramo for the success of this EFSA initiative, where risk analysis has been carried out and dealt with applying the andragogy principles for effective adult training. Subject matter experts and methodological experts worked together, hand-in-hand, throughout the entire learning project lifecycle. This guaranteed the production and delivery of the preparatory eLearning training course on RVF, the development of the case study and relevant training materials and the implementation of the table-top exercise on a dedicated webinar platform (CiscoWebex) according to a well-structured agenda where plenary and sub-group activities have been duly balanced to let the participants feel actors of the capacity building pathway and, finally yet importantly, the satisfaction analysis and the reporting.

Five veterinarians with a solid scientific background in RVF, risk assessment and veterinary epidemiology, two senior pedagogues and one event manager accompanied the evolution of this immersive and blended learning experience.

Participants originated from Italy, Spain, Portugal, France, Greece, Albania, Croatia, Montenegro and Turkey. Twenty-seven participants from 6 EU MSs and 3 EU candidate countries and 12 observers from 3 EU MSs and 3 international organisations attended the training. Participants were enrolled, with the support of the national EFSA Focal Point, from animal health, public health and food safety disciplines as well as per their job role and involvement as risk assessor, risk manager or risk communicator. All participants belonged to public administrations or institutions.

During the simulation exercise, participants were organised into four working groups (WGs) and invited to answer three sets of questions in three different essential steps. The participants were allocated to each WGs to constitute multidisciplinary (including people from the two sectors—veterinary and public health—and three main disciplines: risk assessment, risk management and risk communication) and multi-countries groups.

Specific templates and other guiding documents, such as power points for each set of questions with practical examples and operative suggestions, were developed and distributed progressively to properly address the participants’ needs during the implementation process and guide them towards achieving the expected learning objectives.

In the elaboration of the outcomes, the participants were asked to not refer to specific authorities, institutions or legal acts, but to consider a broader European context.

Supported by the facilitators, the discussion within each WG was conducted to reach consensus-based conclusions, which were then presented by a rapporteur in a subsequent plenary feedback session.

In the first set of questions, the participants were asked to provide their opinions about the disciplines to be represented within the National Crisis Unit (NCU) to coordinate the control measures in the best way.

The second set of questions was asked to identify the main relevant epidemiological data to be collected for a proper assessment of the situation, and the ways the data should be exchanged among administrations. The participants were invited to follow a ‘One Health’ approach in this exercise, considering the importance of data sharing among all administrations for the implementation of integrated control measures.

The relationship with European Institutions was also considered to assure the harmonisation of applied control measures across national borders. Therefore, for the third set of questions, the participants were asked to discuss and elaborate on the possible risk questions to be posed to EFSA, to get useful scientific opinions on the best control and prevention options to be applied, and to identify the other relevant factors potentially guiding the decision of the Animal Health authorities.

The case study and related training tools, as well as the eLearning training course, have been subject to a constant and accurate peer-review and validation coordinated by the EFSA and carried out by a group of experts ad hoc selected within the European Commission (DG Sante), the European Centre for Disease Prevention and Control (ECDC), the Food and Agriculture Organization of the United Nations (FAO), the World Health Organization (WHO) and the World Organisation for Animal Health (WOAH). Representatives of the same Institutions have been involved as observers during the table-top exercise. They contributed to the formulation of the lessons learned and the conclusions given by the Scientific Coordinator who waved the plot of this very articulated simulation exercise.

At the end of the eLearning course, participants completed an evaluation questionnaire made of twenty-two questions. They were encouraged to express their point of view on the eLearning course, the informative presentations and the interactive training sessions. Similarly, the simulation exercise was also evaluated by the participants through a questionnaire composed of 18 questions. Both questionnaires were presented online on a dedicated section of the webinar platform. Summaries of responses were analysed using Microsoft Excel^®^ 2016.

## 3. Results

### 3.1. Outcomes of the Simulation Exercise

As outcomes of the first set of questions, the participants indicated several institutions to be included in the NCU, covering the three risk sectors (risk management—RM; risk assessment—RA; risk communication—RC) (Table 1).

This multisectoral approach was considered particularly relevant by the participants for a vector borne zoonosis such as RVF, as well as the engagement of several different professionals, such as entomologists, climatologists, pests control experts, biologists, epidemiologists, statisticians, social media professionals, economists, in addition to representatives from animal and public health disciplines.

Concerning the second set of questions, the main data identified by the WGs were the following:Data characterising the outbreaks and the susceptible host populations (affected species including humans, clinical signs, breeding locations, number of animals at each location, etc.),Data on the movements of animals and to trace back the origin of animal products,Entomological data related to the locations where mosquito pools were tested and those resulted positive for the presence of RVFV,Results of the serological monitoring in animals and locations where virological positive humans were detected,Data on the production of food of animal origin,Environmental—climatic data.

Figure 1 summarises the stakeholders indicated by the participants, together with the communication strategies (objectives of the messages and possible communication channels) to be put in place for each of them. In general, the contents of the communications suggested by the participants intended to inform stakeholders about the situation, raise awareness and provide instructions to prevent human infection. More specifically, messages to medical professionals and veterinarians also included instructions about the data and information to be collected, whereas those dispatched to consumers and the general public aimed at informing about the actions put in place by the national authorities and providing clear messages about food safety aspects.

Concerning the risk questions to be posed to the EFSA, the participants identified several issues concerning the capacity for the virus to persist (overwintering) and spread (by infected mosquitoes or by animal movements, and relative speed of transmission) within the infected country and to close neighbouring countries, the most probable geographical extension of infection under different spread scenarios, including the effect of weather conditions and mosquito population structure, the relative effectiveness of various measures in reducing the spread of the virus (e.g., use of insecticides vs. vaccination vs. quarantine or stamping out), and the potential role of wildlife in virus transmission in the European context.

Table 2 summarises the participants’ responses regarding the additional factors to be taken into account by the Animal Health Authorities in the decision-making process.

### 3.2. Evaluation of eLearning Course and Simulation Exercise by Participants

Concerning the eLearning course on RVF, 15 participants filled the final evaluation online questionnaire. The general appreciation of the course was high, with an average score of 4.7, considering a scale from 1 (low) to 5 (high) (Figure 2). The quality of the training materials was considered good (average score: 4.5) (Figure 2). The contents of the eLearning training course were considered appropriate and relevant in relation to learning needs (average score: 3.9), although less immediately useful for daily job (average score: 3.7). This probably reflects the general contents of the eLearning course, which aimed to provide the participants with some basic knowledge about the disease.

The simulation exercise was evaluated by the 26 participants, who expressed their evaluation giving scores from 1 to 5. Concerning the general appreciation of the simulation exercise, the average score reported by the participants was 4.4 (scale from 1 = insufficient to 5 = excellent), whereas the usefulness of the event in relation to the daily job reported an average score of 3.6 (scale from 1 = not useful to 5 = very useful) (Figure 2).

## 4. Discussion

The 2021 EFSA/IZS-Teramo Animal Health Crisis Preparedness exercise was originally planned as a ‘physically attended’ event, but the COVID-19 pandemic forced to reconsider and re-plan it as a ‘distance-learning’ event. A combination of methods and resources were taken into consideration to provide effective training.

In order to harmonise the ‘entry knowledge’ of participants, who originated from different institutions/organisations, various countries and diverse professional positions/disciplines, some eLearning modules describing the main features of RVF were made available to the participants one month before.

A table-top simulation exercise was designed as an ‘incident case’. This training approach, based on the presentation and discussion of a particular event, is very appropriate to stimulate the decision-making processes and problem-solving skills, providing the basis for the joint evaluation of alternative solutions.

Each participant brought his/her own relevant competencies, experiences, perspectives and proposals during the working group activities. The facilitators were trained in advance to motivate the engagement of all participants during the discussions, promoting collaboration and a way to seek consensus between members of the working group when synthesising various views into a group opinion. The facilitators have also monitored the set time schedule and stimulated topic contributions from WG members to support progression along the exercise. Pre-defined standardised forms were utilised to structure the discussions and to record and summarise the views.

The WG members enthusiastically engaged in dynamic and fruitful interactions at the simulation exercise, resulting in promising achievement of the established goals.

Coordinating various WG sessions from a ‘control cabin’, using the functions available on the Webex platform, offered a good overarching view of the virtual rooms, plenary sessions or WG compositions and allowed timely technical response. A continuous parallel communication via mobile devices was organised ‘behind the scenes’ between the facilitators, the scientific coordinator, the web platform manager and the eTutor to solve ad hoc tech problems or agree on trigger points for next steps or any necessary adjustments.

The participants accepted well the virtual environment of the exercise, and many demonstrated great confidence in the use of online tools. The WGs appreciated the use of pre-defined templates to guide their discussions.

It is noteworthy that each group demonstrated an original view in its answers, despite the fact that the same questions were posed to all WGs.

All WG outcomes expressed a strong belief in the importance of adopting the ‘One Health’ approach in all steps of RVF crisis management. In addition, the WGs highlighted the importance of a proper management of the crisis preparedness phase. The need for collecting and sharing of data coming from several disciplines (entomology, climatology, wildlife sciences, etc.,) was considered, taking into account the necessary skills, capacities and facilities for analysing the information in a multidisciplinary framework. In addition, emphasis was put on the importance of entomological surveillance and mosquito control, and to establish transboundary surveillance systems and internationally harmonised control strategies, with a proper evaluation of the impact of control measures on the environment and their social acceptability. Finally, given the importance of awareness campaigns for the stakeholders (farmers, slaughterhouse workers, private vets, etc.), the participants draw attention to develop proper communication skills and to have dedicated personnel for effective communication strategies.

The results of the final evaluation of the event were positive, and the participants expressed their appreciation for the activities, despite the limitations caused by the fact that the event was held exclusively online due to restrictions deriving from the COVID-19 pandemic.

Feedback in the online evaluation questionnaire also provided many suggestions on possible further topics to be included in future training workshops, such as for example transboundary diseases, emergency management principles and risk communications.

In particular, some further suggestions were expressed by the participants in relation to the possible future initiatives on the same topic:Simulation exercises should be used more frequently for the identification of gaps in a ‘real context’.After this type of exercise, a follow-up evaluation should be performed, to verify how the exercise contributed to improving the preparedness of countries in relation to a possible RVFV introduction.Larger WGs and/or use of additional facilitation techniques should be considered to encourage contributions from a wider range of participants from several disciplines.To emphasise the discussions within the WGs, providing them as much time as possible for discussions and case studies analyses.To consider capturing WG feedback through collaborative editing tools, other than presentation slides, to make the process more interactive.To schedule consecutive online events over a longer period of days, to better facilitate the concentration of participants during each session.To consider the diversity and specificity of each participating country in a greater extent, taking into consideration specific national circumstances, such as national regulatory frameworks and organisation of decision-making levels, as well as practical case studies from each participant country;To make greater use of pre-read materials/eLearning modules, including contents more closely focused on the management of health emergencies.

It was also suggested to incorporate additional themes in future events, covering the aspects related to preparedness of health and veterinary services in the integrated management of zoonotic emergencies caused by transboundary diseases, following the emergency management principles (e.g., bronze, silver and gold command)—e.g., in reference to the FAO’s ‘Good Emergency Management Practice’ guide [15]. The inclusion of topics concerning the psychological support of personnel involved in crisis management and farmers strongly hit by the emergency was also suggested.

In our opinion, the training event described in this paper represents an interesting example of crisis preparedness exercise, even in a context characterised by difficulties of organising and attending residential training course, as during the COVID-19 pandemic. Although the simulation was performed only virtually as table-top exercise, it allowed the participants to critically discuss the main aspects for a proper crisis management, pursuing a good interaction among all of them. The training approach presented here, therefore, can be considered a good complement to in-field emergency preparedness exercises.

## Figures and Tables

**Figure 1 microorganisms-10-01864-f001:**
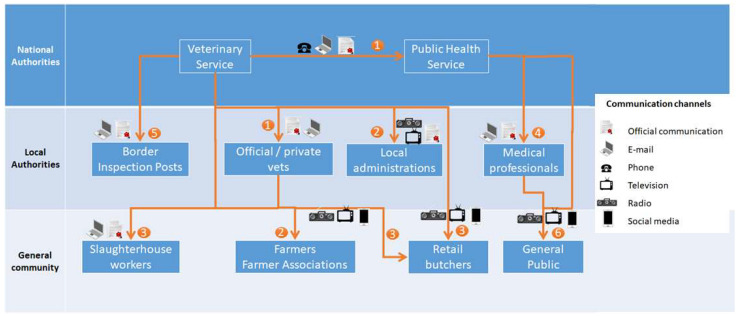
Representation of potential stakeholders and communication channels. The main objectives of the messages are (1) To inform about the case definition, and to indicate the type of data that must be collected. (2) To raise awareness, to advise about the correct behaviours on animal management and the prescriptions issued by the veterinary authority, to reinforce the knowledge about biosecurity measures. (3) To advise about the necessity of following good practices (e.g., PPE use, suspects reporting, proper animal manipulation, etc.,) and the need for caution/protective measures when handling potentially affected animal carcasses. (4) To inform about case definition and clinical best practice, to provide indications on contents of communications with patients, and to indicate the type of data that must be collected. (5) To strengthen controls at the borders. (6) To inform about the situation and the actions put in place by the national authorities, to raise awareness about non-official information.

**Figure 2 microorganisms-10-01864-f002:**
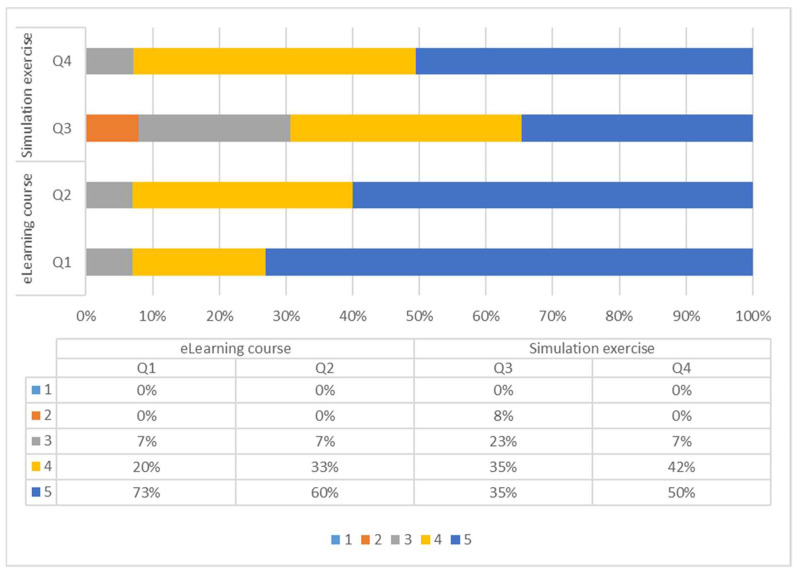
Proportions of the respondents’ scores for the questions concerning the preparatory eLearning course and the simulation exercise: (Q1) What is your general appreciation of the preparatory eLearning course? (scale from 1 = low to 5 = high); (Q2) please evaluate the general quality of the training materials (scale from 1 = low to 5 = high); (Q3) how useful do you consider this event in relation to your daily job? (scale from 1 = not useful to 5 = very useful); (Q4) what is your appreciation of the simulation exercise? (scale from 1 = insufficient to 5 = excellent).

**Table 1 microorganisms-10-01864-t001:** Institutions and risk sectors to be represented in the National Crisis Unit.

Level	Institution	Risk Sector ^1^
National	Ministries of Human/Animal Health	All sectors (RM/RA/RC)
	Ministry of Agriculture	
	Ministry of Environment & Finance Representatives of regions	
	Ministry of Defence/Interior Affairs	
	Veterinary Chamber	
	Veterinary and Public Health Laboratories	
	Ministry of Transport	
	National Crisis Management Committee	RM
EU	European Food Safety Authority	RA
	European Centre for Disease Prevention and Control	
	EU reference laboratories	RM
International	International organisations (e.g., WOAH, EC, EuFMD, WHO)	RM
	Additional lab resources outside country for diagnosis, confirmation etc.	

^1^ RM = risk management; RA = risk assessment; RC = risk communication.

**Table 2 microorganisms-10-01864-t002:** Summary of participants’ responses to the question: ‘Which other factors should guide the decision of Animal Health authorities, in addition to the outcomes of the EFSA risk assessment?’

Description of Factors to Be Considered	Reasons for Considering	Importance for Final Decision (Relevant/Marginal)
Socio-economic considerations on the whole food production chain.	Financial impact on organisations/sectors.	Relevant to consider the whole chain (e.g., from farmers to final manufacturers and consumers), but taking into account also the impact on associated areas, such as, for example, tourism.
Public perception of ‘severe’ control measures (e.g., animal mass culling).	Political/societal acceptance of risk management actions.	Relevant insofar as more controversial measures may provoke contentious reactions, thus hampering their proper implementation.
Environment factors (e.g., use of insecticides).	Public concern due to zoonotic characteristics of the disease (and implications for food consumption habits).	Relevant to find a compromise between the need for activating urgent control measures and public concern for the environmental impact of the use of certain insect control strategies.
Impact on human health/Economy/Social impact	Impact on public health and ecosystems from extended/extensive use of insecticides.	Relevant—when considering insecticides as one of the control measure options. Cost-benefit analysis of control options should be considered.
	Human disease.	Very Relevant due to the severe impact on public health potentially deriving.
Vulnerable human category (economic/social impact)	Economic impact of stamping out policy and that deriving from the restrictions to international trade.	Relevant when a substantial part of the economy is involved and/or a high number of farmers or food producers are impacted.

## Data Availability

Not applicable.

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
