# Peer review of "Crisis Preparedness Exercise on Rift Valley Fever Introduction into Europe under a One Health Approach"

_microorganisms, 2022, doi:10.3390/microorganisms10091864_

Round 1
Reviewer 1 Report
The manuscript entitled „Preparedness of Rift Valley Fever introduction into Europe un- 2 der a One Health approach” by Pediconi et al. describes the methodology and outcome of a crisis preparedness training program, in this case specifically for RVFV, but with the potential to be broadened to other contagious diseases.
The content of the manuscript describes an important aspect of one health approach and prevention of infectious disease and their spread throughout the country/continents. However, the manuscript comes along with some weaknesses that makes it harder for the reader to follow the authors reports and partially lacks a clear line of thoughts.
Specific comments:
Although English language is already at a good, al most native level, the whole manuscript should be double checked another time. Some minor mistakes in grammar or spelling can be found throughout the manuscript.
The manuscript consist to almost 20% of different bullet points. This leaves the impression of rather a enumerative than a narrative manuscript. Consider to change at least part of the many sections with bullet points to either a figure or a narrative text.
Reading the results sections, it is very hard to follow what the section is describing. To make it easier for the reader to follow the line of thoughts, please include subheadings that mirror the materials and methods section.
Table 2: Would it also be possible to transfer the information of this table to a figure and to also visualize how different stakeholders and tasks are interconnected? That would be a very helpful tool
Throughout the whole manuscript: please double check the use of RVF vs. RVFV for the disease vs. the virus.
Figure 1 is not sharp and a bit blurry
Figure 2: Although the same content/message is visualized as in figure 1, the figure looks completely different. It would be reader friendly to keep both figures consistent with each other.
ll.268-282 are rather fitting into the materials and methods section
l.363: So what is the conclusion? So far the manuscript is rather descriptive and lacks interpretations and a clear conclusion
After reading the whole manuscript, it is a bit unclear if the authors want to focus on the description of the training program or the scientific and specific outcomes of it. Either put a clear focus on the passages of the text or divide it with subheadings or sub-sections
The title “Preparedness of Rift Valley Fever introduction into Europe under a One Health approach” is promising another content than that that is presented in this manuscript. With the presented work Europe is not prepared for a RVFV introduction, but a preparedness program to get there is presented. Please consider to revise the title, so that the text will meet the readers expectations.
Author Response
We thank the reviewer for the improving suggestions given and we hope to have properly addressed the comments received. Below our answers to each comment.
Specific comments:
Although English language is already at a good, almost native level, the whole manuscript should be double checked another time. Some minor mistakes in grammar or spelling can be found throughout the manuscript.
A more in depth English revision has been performed.
The manuscript consist to almost 20% of different bullet points. This leaves the impression of rather a enumerative than a narrative manuscript. Consider to change at least part of the many sections with bullet points to either a figure or a narrative text.
Bullet points in Results and Discussion have been translated into narrative text.
Reading the results sections, it is very hard to follow what the section is describing. To make it easier for the reader to follow the line of thoughts, please include subheadings that mirror the materials and methods section.
The results chapter has been divided into two sub-chapters. The text on the evaluation of events by participants has been improved, as well as the Figures.
Table 2: Would it also be possible to transfer the information of this table to a figure and to also visualize how different stakeholders and tasks are interconnected? That would be a very helpful tool
Table 2 has been completely converted into a Figure. More explanations have been included in the text.
Throughout the whole manuscript: please double check the use of RVF vs. RVFV for the disease vs. the virus.
The use of these acronyms has been checked. Thanks
Figure 1 is not sharp and a bit blurry
Figure 2: Although the same content/message is visualized as in figure 1, the figure looks completely different. It would be reader friendly to keep both figures consistent with each other.
Previous Figure 1 and Figure 2 have been merged into a unique representation. We hope that this is now clearer and concise.
ll.268-282 are rather fitting into the materials and methods section
Sorry but we were not able to clearly identify the lines. At the lines 268-282 the outcomes of working groups are discussed. Maybe the reviewer referred to other lines?
l.363: So what is the conclusion? So far the manuscript is rather descriptive and lacks interpretations and a clear conclusion
More clear and explicit conclusions have been added at the end. We hope that this can clarify.
After reading the whole manuscript, it is a bit unclear if the authors want to focus on the description of the training program or the scientific and specific outcomes of it. Either put a clear focus on the passages of the text or divide it with subheadings or sub-sections
We want to focus more on the training approaches used, considering the limitations posed by the COVID-19 pandemic. In our opinion this experience represents a good example for possible alternative or complementary approach for promoting the crisis preparedness in the countries.
The title “Preparedness of Rift Valley Fever introduction into Europe under a One Health approach” is promising another content than that that is presented in this manuscript. With the presented work Europe is not prepared for a RVFV introduction, but a preparedness program to get there is presented. Please consider to revise the title, so that the text will meet the readers expectations.
We agree with the reviewer. The title has been modified as “Crisis preparedness exercise on Rift Valley Fever introduction into Europe under a One Health approach”
Reviewer 2 Report
Dear Authors
Thank you for the manuscript. It was a pleasure reading it. I have attached the PDF document with my comments in it. Basically, I felt that the readers could get more from the paper if the programmes used to analyse the data were provided.
Also, it would help if we knew how each of the 4 WGs were constituted e.g by the participant's professions, and to see if that affected the kind of suggestions they made; if the non-veterinary groups took more time to understand and whether they would require more refresher courses etc.
In addition, did the presence of people in various government authorities (if present) help or not. Your paper extensively analysed what the course attendants said and did, but did not analyse your assessment of them and their answers, given their backgrounds and living in a RVF free country.

Author Response
Dear Authors
Thank you for the manuscript. It was a pleasure reading it. I have attached the PDF document with my comments in it.
Dear reviewer, we want to thank you for the useful suggestions. Suggested changes have been reported in the revised text.
Basically, I felt that the readers could get more from the paper if the programmes used to analyse the data were provided.
The questionnaires were presented online on a dedicated section of the webinar platform, where the answers of respondents are recorded. Summaries of responses were analysed using Microsoft Excel®. This clarification has been added to the Materials and methods.
Also, it would help if we knew how each of the 4 WGs were constituted e.g by the participant's professions, and to see if that affected the kind of suggestions they made; if the non-veterinary groups took more time to understand and whether they would require more refresher courses etc.
We added more details in the Materials and methods, explaining that we mixed people coming from different disciplines and sectors (veterinarians and public health sectors) and from different countries in order to create heterogeneous groups, in which all different origins and expertise were represented. Therefore no differences in the approaches or answers of WGs were observed.
In addition, did the presence of people in various government authorities (if present) help or not. Your paper extensively analysed what the course attendants said and did, but did not analyse your assessment of them and their answers, given their backgrounds and living in a RVF free country.
It was not possible to look at single individual differences, because to outcomes were not individual but for each WG, In fact, the participants of each WG discussed together till they found a consensus and the outcomes were expressed as group’s work. In addition, the WG were constituted mixing as much as possible people coming from different countries, sectors and disciplines.